# The Roles of eIF4G2 in Leaky Scanning and Reinitiation on the Human Dual-Coding POLG mRNA

**DOI:** 10.3390/ijms242417149

**Published:** 2023-12-05

**Authors:** Ekaterina D. Shestakova, Roman S. Tumbinsky, Dmitri E. Andreev, Fedor N. Rozov, Ivan N. Shatsky, Ilya M. Terenin

**Affiliations:** 1Faculty of Bioengineering and Bioinformatics, Lomonosov Moscow State University, 119234 Moscow, Russiatu_roman2018@mail.ru (R.S.T.); 2Belozersky Institute of Physico-Chemical Biology, Lomonosov Moscow State University, 119234 Moscow, Russiashatsky@belozersky.msu.ru (I.N.S.); 3Shemyakin-Ovchinnikov Institute of Bioorganic Chemistry, RAS, 117997 Moscow, Russia; 4Department of Biochemistry, School of Biology, Lomonosov Moscow State University, 119234 Moscow, Russia; 5Translational Medicine Research Center, Sirius University of Science and Technology, Olimpiyskiy ave. b.1, 354349 Sochi, Russia

**Keywords:** translation reinitiation, 40S, mitochondrial disfunction, polyglutamine, ribosome collision, cap-dependent translation, non-AUG translation, MYCBP2, PHD2, AUG selection

## Abstract

Upstream open reading frames (uORFs) are a frequent feature of eukaryotic mRNAs. Upstream ORFs govern main ORF translation in a variety of ways, but, in a nutshell, they either filter out scanning ribosomes or allow downstream translation initiation via leaky scanning or reinitiation. Previous reports concurred that eIF4G2, a long-known but insufficiently studied eIF4G1 homologue, can rescue the downstream translation, but disagreed on whether it is leaky scanning or reinitiation that eIF4G2 promotes. Here, we investigated a unique human mRNA that encodes two highly conserved proteins (POLGARF with unknown function and POLG, the catalytic subunit of the mitochondrial DNA polymerase) in overlapping reading frames downstream of a regulatory uORF. We show that the uORF renders the translation of both POLGARF and POLG mRNAs reliant on eIF4G2. Mechanistically, eIF4G2 enhances both leaky scanning and reinitiation, and it appears that ribosomes can acquire eIF4G2 during the early steps of reinitiation. This emphasizes the role of eIF4G2 as a multifunctional scanning guardian that replaces eIF4G1 to facilitate ribosome movement but not ribosome attachment to an mRNA.

## 1. Introduction

Translation initiation in eukaryotes begins in most cases with an attachment of the small ribosomal subunit to the 5′ m^7^G-cap via the eIF4F, a multifunctional heterotrimeric complex composed of eIF4A, eIF4E, and either eIF4G1 or eIF4G3 [1,2,3,4]. During the initiation cycle, the eIF4F engages in multiple activities. First, it binds to the cap via its eIF4E subunit to ensure the anchorage to the mRNA. Second, the scaffold eIF4G1 binds eIF3, which in turn binds the 40S ribosomal subunit, thereby linking the mRNA to the ribosome. Third, the eIF4G1 binds and invigorates the eIF4A, otherwise a non-processive RNA helicase, to melt mRNA secondary structures during the initial ribosome accommodation and subsequent scanning. Fourth, eIF4G1 binds the polyadenylate-binding protein (PABP) to bend the mRNA in a closed loop. All of these activities can be regulated, thereby making eIF4F an important lever for translational control.

Mammalian genomes encode a few proteins that are homologous to eIF4G1. One of them, eIF4G2 (also known as DAP5, Nat1, or p97), only partially fulfills the functions of eIF4G1, as it lacks the PABP and eIF4E binding sites and, unlike the eIF4G1, binds only one molecule of eIF4A, whereas eIF4G1 binds two [5,6,7]. Another seemingly consequential difference is that, unlike eIF4G1, eIF4G2 binds to eIF2 [8,9,10]. Although the relevance of this interaction is unknown, eIF4G2 that is defective in the eIF2 binding fails to function [11].

eIF4G2 generally promotes translation of mRNAs with long 5′ leaders and uORFs [12,13,14], although a few mRNAs lacking uORFs also require eIF4G2 [12,15]. Mechanistically, eIF4G2 is thought to participate in reinitiation after the uORF translation [12] or to substitute for eIF4G1 to promote leaky scanning through a translated uORF when ribosomes that have leaked through an uAUG interfere with ribosomes that translate the uORF, lose the eIF4G1, and fail to scan further, unless they reacquire the helicase module via the eIF4G1 or eIF4G2 [5,13]. According to either model, eIF4G2 can compensate for an eIF4G1 deficiency in scanning but not in ribosome attachment. Which of the two divergent processes the eIF4G2 bolsters, or whether it does indeed ensure both, is currently unclear.

Human or murine embryonic cells lacking eIF4G2 fail to differentiate properly [9,16,17]. In mice, deficiency in Map3k3, which is the eIF4G2 mRNA target [9,13,15], has been suggested to be the major determinant of mES’s inability to differentiate [9]. On the other hand, *eIF4G2* knockout also impairs respiratory complex I activity in the hES cells, whereas blocking oxidative respiration prevents the cells from retinoic-acid-induced differentiation [16].

The mRNA for the catalytic subunit of the mitochondrial DNA polymerase, POLG (sometimes referred to as POLG1), is a rare case of a genuine bicistronic mRNA in mammals (Figure 1A). In addition to POLG, it encodes a highly conserved protein POLGARF (POLG Alternative Reading Frame) of unknown function in an overlapping CUG-initiated reading frame [18,19]. Translation of both POLGARF and POLG is controlled by an uORF, and, consequently, both leaky scanning and reinitiation contribute to their expression [18] (schematically shown in Figure 1B,C).

*POLG* is one of the few nuclear encoded genes associated with mtDNA disorders [21]. It bears a polyglutamine tract, and this tract’s expansion has been linked to Parkinson’s disease and other neuropathological conditions [22,23]. The complex pattern of the ORFs in the POLG mRNA and its physiological importance prompted us to test whether eIF4G2 is involved in its translation. Here, we show that eIF4G2 indeed promotes both POLGARF and POLG translation, and we dissect the eIF4G2’s role in both leaky scanning and translation reinitiation on a single mRNA.

## 2. Results

### 2.1. eIF4G2 Participates in POLG and POLGARF Translation

First, we tested if eIF4G2 is required for either POLG or POLGARF translation. The POLG and POLGARF reporter constructs were cloned previously [18]. We transfected in vitro transcribed m^7^G-capped and polyadenylated reporter mRNAs into cells depleted of eIF4G2 and, in accordance with our expectations, the POLG translation is significantly eIF4G2 dependent, while the POLGARF translation is as well, to a bit of a lesser extent (Figure 2A).

Among others, reporters for PHD2 (prolyl hydroxylase domain-containing protein 2), the expression of which has been demonstrated to decrease upon eIF4G2 depletion [11], and for MYCBP2 (MYC Binding Protein 2), which was identified as an eIF4G2 target through ribosome footprint profiling [12], also proved to be eIF4G2 dependent (Figure 2A and Appendix A).

Several reports have suggested that eIF4G2 can operate in a separate mechanism, where the cap-binding is executed by eIF3d and then the direct eIF3d-eIF4G2 interaction drives the scanning complex accommodation [24,25,26]. First, we treated cells with the mTOR inhibitor PP242 in order to determine whether the translation of POLG and POLGARF is eIF4E dependent. We found that 1 uM of PP242 inhibited the POLG and POLGARF translation in a way that was comparable or even stronger than that of other cap-dependent reporters, thus showing that eIF4E is the major cap-binding protein for the POLG mRNA (Figure A1A). Second, knocking down eIF3d did not affect the POLG or POLGARF translation (Figure A2). This demonstrates that the POLG mRNA translation is primarily driven by eIF4E rather than eIF3d [26].

### 2.2. The uORF Determines the Reliance of the POLG and POLGARF Translation on eIF4G2

In the majority of previously dissected cases, the presence of uORF(s) correlated with the requirement for eIF4G2 for translation, and elimination of the uORFs diminished the need for the protein [12,13,16,27]. We investigated whether this was true for the POLG or POLGARF translation. Indeed, the uORF elimination rendered the translation of both reporters insensitive to the eIF4G2 depletion (Figure 3, Figure A3 and Appendix A). However, several issues make the POLGARF/POLG case more complex. To begin with, a recent study found that eIF3d or eIF4G2 depletion can specifically improve translation from a CUG start codon [28]. We failed to observe increased POLGARF translation upon eIF3d depletion (Figure A2A), nor did the CUG to AUG mutation change the POLGARF susceptibility to eIF4G2 depletion (Figure 3B and Figure A3B). Second, the downstream stem-loop is required for efficient non-AUG translation initiation on the POLGARF start codon [18]. Its disruption via silent mutations increased the requirement for eIF4G2 of POLGARF while decreasing it for POLG (Figure 3 and Figure A3). The stem-loop disruption arguably impairs the POLGARF translation, and, thus, the scanning ribosomes encounter fewer impediments on their way to the POLG start codon. As a result, the POLG translation indeed would become less reliant on eIF4G2. That seems not to be the case, though, and we return to this later.

### 2.3. The Roles of the Upstream and the POLGARF Start Codons in POLG Translation

Because the initial report focused primarily on the POLGARF translation [18], we separately mutated the uAUG or the CUG start codon to better understand how the upstream and POLGARF ORFs control the POLG expression (Figure 4). The elimination of the uORF severely inhibited POLG translation (Figure 4A and Appendix A), whereas the stem-loop disruption reversed this effect.

The outcomes of the stem-loop disruption on both POLGARF and POLG translation corroborate the notion that it is required for the efficient POLGARF translation. For example, the CUG to AUG mutation only marginally (~1,5-fold) augmented the POLGARF translation when the stem-loop was intact, but did it strongly (~4-fold) augment the POLGARF translation when the stem-loop was mutated. Interestingly, the CUG to AUG mutation virtually precluded the POLG translation (~14-fold decrease) irrespective of the stem-loop integrity, suggesting a non-linear relation of a start codon strength and its apparent leakiness (see Section 3).

### 2.4. A Contribution of eIF4G2 to Reinitiation and Leaky Scanning in POLG and POLGARF Translation

Because there is an evident discrepancy as to whether eIF4G2 contributes to leaky scanning [13] or reinitiation [12], we investigated what type of translation eIF4G2 promotes in the case of POLGARF and POLG. For this purpose, we mutated the uORF stop codon so that the extended uORF now overlapped out of frame with the POLGARF ORF (referred to as “POLGARF uORF ext”) (Figure 5D and Appendix A). As a result, the POLGARF translation became completely dependent on the ribosomes that leak through the uAUG (Appendix A). Alternatively, we fused the uORF to the POLGARF frame so that the POLG translation was now solely dependent of the leaky scanning (referred to as “POLG uORF ext”) (Figure 5C and Appendix A). In a complementary approach, we introduced two in-frame AUG codons in good contexts into the uORF to filter out the leaky ribosomes and make the corresponding POLGARF and POLG reporters primarily dependent on the reinitiation (referred to as “uAUG x2”) (Figure 5C,D and Appendix A). Consistent with the previous observations [18], the POLGARF translation is initiated by roughly equal amounts of the leaky and reinitiating ribosomes, whereas a major fraction of the POLG translation is performed by the reinitiating ribosomes (Figure 5E and Appendix A). The dual mutation that prevented both leaky scanning and reinitiation greatly reduced the translation to an almost background level (Figure 5E and Appendix A), demonstrating that the introduced uAUG codons do strongly suppress the leaky scanning and that nearly all of the Fluc translation on the “mostly reinitiation” reporters is indeed mediated by the reinitiation. The suppression of leaky scanning made the POLG translation more dependent on eIF4G2 and, reciprocally, the elimination of reinitiation made it slightly less susceptible to eIF4G2 depletion (Figure 5A,B, Figure A4 and Appendix A). Therefore, eIF4G2 secures the POLG translation by promoting both leaky scanning and translation reinitiation.

Notably, while the stem-loop disruption had only a minor effect on the POLG translation that stems from reinitiation, it significantly enhanced the POLG translation that relied on leaky scanning (Figure 6A). It could be argued that the leaky ribosomes are more sensitive to the secondary structure than reinitiating complexes. However, a mutually non-exclusive explanation is that most of the reinitiating ribosomes bypass the POLGARF start because it is too close to the uORF stop codon, whereas effective reinitiation necessitates a sufficient intercistronic length [29,30,31,32]. The observations that a) the stem-loop disruption affects both modes of POLGARF initiation roughly equally (Figure 6B); and (b) the CUG mutant variant of the POLG reporter does not respond to the stem-loop mutation (Figure 6A) indirectly support this second interpretation.

Reinitiation is a poorly understood process [33,34,35,36] that includes (a) the transition from the post-termination state to the scanning-competent state, (b) the reacquisition of eIF2, and (c) the scanning itself. The reinitiated scanning ribosomes can potentially interfere with ribosomes translating a downstream ORF (POLGARF in our case), as they do in the case of eIF4G2-dependent leaky scanning. While the readout would seem to suggest that eIF4G2 promotes reinitiation, in fact this would not exclude the possibility that eIF4G1 is a major factor in the initial steps of reinitiation [37,38], while eIF4G2 predominantly contributes to subsequent scanning. Thus, we eliminated the CUG POLGARF start codon from the “mostly reinitiation” POLG reporter and investigated whether this changed the requirement for eIF4G2. It did not (Figure 5A, Figure A4A and Appendix A).

At the same time, the “mostly reinitiation” POLGARF reporter translation remained eIF4G2 dependent (Figure 5B, Figure A4B and Appendix A), regardless of the presence of the stem-loop. The short distance between the uORF stop and the POLGARF start codons (only 11 nt) arguably gives eIF4G2 no chance to replace the eIF4G1. We have to suggest, therefore, that eIF4G2 is present in the reinitiating complexes right from the beginning, and thus the protein promotes the genuine translation reinitiation in the case of both POLGARF and POLG. Notably, the effects of the CUG mutation in the wild-type and “mostly reinitiation” reporters demonstrate that the leaky ribosomes mainly initiate at the POLGARF start, whereas the reinitiated ribosomes mainly bypass the CUG to initiate downstream at the POLG start (Figure 5F and Appendix A).

### 2.5. Reapprasial of the eIF4G2 Contribution to the Translation of the Maf1, Stard7, and UCP2 mRNAs

The successful distinction between the leaky scanning and reinitiation modes prompted us to ask the same question about the previously identified eIF4G2 mRNA targets, i.e., Maf1, Stard7, and UCP2. We have previously demonstrated that eIF4G2 promotes leaky scanning on these mRNAs [13], but we were unable to contemplate the reinitiation due to the experimental design. Thus, analogous to the POLG case above, we introduced two in-frame uAUG codons in good contexts into either the wild-type or the extended uORFs (overlapping with Fluc sequenced) of the Maf1, Stard7, and UCP2 reporters (Figure 7C,D). Again, the double modification almost eliminated the Fluc expression (Figure 7E and Appendix A), and all three uORFs arguably tend to control the downstream translation via both mechanisms (Figure 7E and Appendix A). Also, similarly to the POLG case, both are sensitive to the eIF4G2 depletion (Figure 7A,B and Appendix A). Notably, the contributions of reinitiation and leaky scanning are different for the mRNAs tested, and the eIF4G2 contribution to the studied mechanisms is also different, all which results in different measurable outcomes of the eIF4G2 depletion.

Therefore, eIF4G2 can promote both leaky scanning and reinitiation on the same mRNA.

### 2.6. Translation of Both POLGARF and POLG Is Sensitive to eIF1 In Vitro

The complex pattern of the POLG mRNA start sites could have suggested the existence of an unconventional regulation by start codon context sensors. We took advantage of in vitro translation, which is unlikely to be affected by the inevitable side effects of eIF1 overexpression in cultured cells.

The translation of the β-globin reporter with a very good context hardly responded to the addition of eIF1, whereas the GUG-initiated eIF4G2 reporter and the wild-type context eIF1 reporter [39] were highly sensitive to eIF1, even at the lowest concentrations tested (Figure 8C). Contrary to the eIF1 overexpression in cells [18], translation from the POLGARF CUG codon was sensitive to eIF1, but it became insensitive upon mutation to AUG (Figure 8B). The POLG translation was sensitive to the addition of eIF1, indicating that the POLG AUG context is not perfect, either (Figure 8A). We thus applied the initiation site score developed by Noderer and colleagues [40] for the start sites tested here. According to these data, the good Kozak initiation context receives a reference score of 100. Only the β-globin start site out of the tested reporters has a score higher than 100 (112), while the other sites were predicted to be less efficient (70–80), and the eIF1 start has a score of only 44. Arguably, the strength of a codon and how eIF1 affects its recognition are linked in a more complex way than anticipated.

## 3. Discussion

Given the predominance of cap-dependent translation in eukaryotes, the potential for efficient translation of two proteins from a dual-coding mRNA is extremely limited. In the case of the POLG/POLGARF mRNA, it is the interplay between the uORF, the relatively efficient CUG (but not AUG) start codon, the POLGARF stem-loop, and eIF4G2 that allows for adequate translation of both proteins.

### 3.1. The Significance of CUG Codon Strengh for POLG Translation

Counterintuitively, the rather small impact of the CUG to AUG mutation on the POLGARF translation (~1,5-fold increase) coincides with the severe inhibition (~14-fold) of the POLG translation (Figure 4A and Appendix A), regardless of whether the stem-loop was intact or not. This phenomenon is not unprecedented. The P/C mRNA of certain paramyxoviruses encodes an N-terminally extended C’ protein, which is effectively initiated at a non-AUG codon. Its mutation to AUG only slightly improves the C’ expression, but it strongly inhibits the downstream initiation at C and P start codons [41,42]. In fact, this is quite easy to comprehend. Assume that half of the scanning ribosomes skip a start codon to initiate translation downstream. Doubling the probability of initiation at the first start essentially nullifies the downstream initiation. Regardless of the underlying mechanism, the data presented demonstrate why POLGARF translation is initiated at the conserved non-AUG codon; otherwise, the expression level of the POLG would be extremely low.

### 3.2. uORF Provides POLG Translation

The uORF is also critical for the translation of both POLGARF and POLG. Substitution of the uAUG for the stop codon promotes initiation at the subsequent POLGARF start codon and strongly inhibits translation at the main POLG start (Figure 4 and Appendix A). It redistributes the reinitiating ribosomes to the downstream POLG start. This phenomenon is not uncommon in human mRNAs, but it has never been generalized. When a short uORF precedes two start codons, it seems to augment translation at the downstream one and attenuate translation at the upstream start, as has been demonstrated for ATF4 [43,44], ATF5 [45,46], SCL [47], PTEN [48], C/EBP⍺, and C/EBPβ [27,49], to name a few. The efficiency of reinitiation after uORF translation depends on the distance between the uORF stop and the downstream AUG [29,30,31,32], and it is thought to be influenced by the availability of the ternary Met-tRNA_i_^Met^-eIF2-GTP complex, i.e., the longer the reinitiating complex scans, the more likely it acquires the ternary complex. However, such redistribution occurs under normal conditions as well and does not require the eIF2 inactivation [27,43,44,45,46,47,48,49], probably hinting that eIF2-independet scanning is an unappreciated yet widespread process [50]. In line with this notion, the POLG reporter showed no resistance to the eIF2 inactivation to thapsigargin treatment despite a clear similarity in the ATF4 and POLG uORF patterns (Figure A1B).

### 3.3. eIF4G2 Role in Leaky Scanning and Reinitiation on a Single mRNA

Previous reports have shown that eIF4G2 can participate in either leaky scanning [13] or reinitiation [12]. We were able to address the contributions of these two mechanisms to the translation of several mRNAs and to differentially estimate their dependence on the eIF4G2. We demonstrate that eIF4G2 can promote both mechanisms on a single mRNA. Reinitiation per se does not require eIF4G [51], so it is quite plausible that either eIF4G1 or eIF4G2 can participate in the subsequent scanning. Our data also emphasize that leaky scanning and reinitiation are not mutually exclusive and can well occur on a single mRNA with different input from each mechanism (Figure 5E, Figure 7E and Appendix A). We elaborate a proposed model [5] of eIF4G2′s involvement in canonical cap-dependent translation on an mRNA with an uORF (Figure 9).

Nevertheless, it remains unclear why POLG translation depends more on eIF4G2 than POLGARF does and why the stem-loop disruption tends to equalize the eIF4G2 contributions to the translation of both proteins. It seems that leaky scanning through the POLGARF non-overlapping part does not contribute to the reliance of the POLG translation on eIF4G2 (Figure 5F and Appendix A). All of this can be explained if the premise that reinitiating ribosomes rely more heavily on eIF4G2 than leaky ribosomes is correct, which appears to be the case, at least for POLG (Figure 5A, Figure A4A and Appendix A). In this case, the more translation initiation depends on reinitiation, the more it depends on eIF4G2. The presented data nicely support the notion that the reinitiating ribosomes mostly ignore the POLGARF start codon due to the very short distance between the uORF stop and the CUG start codons and, conversely, we see that the POLG translation is mostly initiated by the reinitiating ribosomes. The stem-loop disruption does not markedly affect the reinitiation, but it promotes leaky scanning through the CUG, thereby increasing the number of less eIF4G2-dependent “leaky” ribosomes that reach the POLG start.

Although substantial progress has been achieved in understanding the eIF4G2 mechanism of action, it is still unclear why certain uORF-containing mRNAs require eIF4G2, while the others do not. For example, another eIF4G2 mRNA target, C/EBP⍺ [12], follows the POLG pattern with an uORF and two start sites. The only difference is that the two downstream codons are in frame, resulting in the translation of two C/EPB⍺ isoforms, rather than two different proteins. This mRNA requires eIF4G2 for the translation of the shorter, but not the longer, isoform [27].

## 4. Materials and Methods

### 4.1. Antibodies and Western Blotting

Antibodies against eIF4G2 (A302-249A), GAPDH (A300-639A), and eIF3d (A301-759A) were purchased from Bethyl laboratories (Montgomery, TX, USA). The HRP-conjugated secondary antibodies were from Invitrogen (anti-rabbit 31460).

Nitrocellulose membranes (0.2 μm, Bio-Rad, Hercules, CA, USA) were blocked in 3% ECL™ Blocking Agent (GE Healthcare, Chicago, IL, USA) in TBST at room temperature for 1 h, probed with antibodies against eIF4G2 (1:5000), GADPH (1:5000), or eIF3d (1:5000), and then detected through chemiluminescence using corresponding anti-rabbit antibodies at a 1:25,000 dilution. Incubation with primary and secondary antibodies was also performed in 3% blocking reagent in TBST under the same conditions. Antibodies bound to eIF4G2, GADPH, or eIF3d were visualized with an enhanced chemiluminescence detection kit (ECL™ Prime Western Blotting System, GE Healthcare, Chicago, IL, USA). The images were captured using ChemiDoc XRS+ with Image Lab™ 3.0 software for image processing and quantification (Bio-Rad, Hercules, CA, USA).

### 4.2. Plasmids

Plasmids encoding either the wild-type POLG and POLGARF reporters or the uAUG or CUG mutants have been described [18]. The stem-loop was disrupted by point mutations, which did not alter the POLGARF amino acid sequence. Plasmid for the expression of eIF1 has been described [52]. PHD2 and MYCBP2 5′ UTRs were amplified from the RKO-derived cDNA. Other reporters have been also described [13]. Oligos for mutagenesis were synthesized by Evrogen (Moscow, Russia), and all plasmids were sequenced there, too.

### 4.3. Transcription

Templates for transcription were generated via PCR that introduced a 50 nt long poly(A)-tail as described [15] and purified using Monarch PCR and a DNA Cleanup kit (NEB). In vitro transcription was performed in a buffer containing 40 mM of DTT, 2 mM of spermidine, 80 mM of HEPES-KOH pH 7.5, and 24 mM of MgCl_2_. The reaction mixture also contained 3 mM each of NTP (Biosan, Novosibirsk, Russia), 12 mM of ARCA cap analogue (Biolabmix, Novosibirsk, Russia), and, per 10 µL of a reaction, 4 units of RiboCare ribonuclease inhibitor (Evrogen, Moscow, Russia), 50 units of T7 RNA polymerase (Biolabmix, Novosibirsk, Russia), 0.4–0.5 μg of the template, and 0.1U of *E. coli* inorganic pyrophosphatase (NEB). The reaction was carried out for 2 h at 37 °C, and another 3 mM of each NTP was added to the reaction and incubated for another 2 h. DNA was hydrolyzed using RQ1 nuclease (Promega, Madison, WI, USA), and RNA was precipitated by 2,8 M LiCl on ice for one hour. The solution was then centrifuged for 15 min (25,000× *g*, 4 °C). The RNA pellet was washed with 70% ethanol and dissolved in nuclease-free water (Evrogen, Moscow, Russia). RNA concentration was determined spectrophotometrically through absorbance at 260 nm.

### 4.4. Transfection

293T and Huh7 cells were cultured under standard conditions in DMEM (PanEco, Russia) supplemented with 10% FBS (HyClone, Cytiva, Marlborough, MA, USA). The duplexes for the eIF4G2 and eIF3d knockdown have been described [13] (see Appendix A (capital letters stand for the unmodified ribonucleotides, and lowercase letters denote the 2′-O-Me protected ribonucleotides)). siRNAs were purchased from Genterra (Moscow, Russia). The eIF4G2 and eIF3d knockdowns in 293T and Huh7 cells were performed as described [13,15]. Briefly, cells were plated in a 4- or 12-well plate (depending on the number of experimental points) at ~25% density simultaneously with the first round of siRNA transfection. The siRNAs (with a final concentration of 10 nM in medium) were transfected using Lipofectamine RNAiMAX (Invitrogen, Thermo Fisher Scientific, Waltham, MA, USA), as suggested by the manufacturer. On day 3 (after 48 h), the cells were replated to a 48-well plate at density ~30% simultaneously with the second round of siRNA transfection. On day 4 (roughly 72 h after the first siRNA application), mRNA transfection was performed. For a well of a 48-well plate, 50 ng of reporter mRNA was mixed with 5 ng of reference mRNA (in vitro transcribed m^7^G-capped and polyadenylated β-globin-Nluc) in 25 μL of PBS. Then, 0.2 μL of GenJect40 reagent (Molecta, Moscow, Russia) was diluted in 25 μL of PBS and incubated at room temperature for 10 min. The volumes were multiplied in accordance with the required number of experimental points. Then, the mRNA mixture was added to the transfection reagent solution, incubated for 15 min at room temperature, and then applied to the cells. Where indicated, cells were treated with 1 μM PP242 (Tocris Bioscience, Bio-Techne SAS, Noyal Châtillon sur Seiche, France) or 1 μM thapsigargin (MilliporeSigma, Burlington, MA, USA) ten minutes prior to transfection. Four (two in case of the PP242 or thapsigargin treatments) hours later, the cells were processed with the Firefly & Renilla Luciferase Single Tube Assay Kit (Biotium, Fremont, CA, USA) according to the manufacturer’s protocol. The luciferases’ activities were measured manually using the Modulus luminometer (Turner Biosystems, Promega, Madison, WI, USA). Normalized reporter expression was calculated by dividing reporter Fluc activity to reference Nluc activity. The knockdown effect was calculated by dividing normalized reporter expression in depleted cells by normalized expression in control cells. The effects of described mutations on translation efficiencies were calculated according to the normalized reporter expression of an mRNA with mutant 5′UTR to the corresponding mRNA with wild-type 5′ UTR, and it is referred to as relative translation efficiency. The effect of drug treatment was calculated by dividing the normalized reporter expression in cells treated with the drug to that in vehicle-treated cells.

### 4.5. Protein Expression and Purification

eIF1 was expressed in Rosetta-gami^TM^ cells (Novagen, Madison, WI, USA) at 30 °C and purified at Ni-NTA agarose (QIAGEN Sciences, Germantown, MD, USA) and then at heparin-sepharose, as described [53].

### 4.6. Preparation of S20 Translation Extract

Translation extracts were prepared using Expi293F (Gibco, Thermo Fisher Scientific, Waltham, MA, USA) suspension cells rather than 293T adherent cells because a significant amount of cells can be easily grown by culturing in Expi293TM Expression Medium (Gibco, Thermo Fisher Scientific, Waltham, MA, USA). Cytoplasmic extracts were prepared as described [54] with minor modification. In total, 6 billion Expi293F cells (100% viability) were collected through centrifugation (300× *g*, 10 min, 4 °C), washed with ice-cold PBS, resuspended in 10 mL of PBS, and collected again (300× *g*, 10 min, 4 °C). Then, the cells were rapidly resuspended in 1 mL lysolecithin buffer (20 mM of HEPES-KOH pH 7.5, 100 mM of KOAc, 2.2 mM of Mg(OAc)_2_, 2 mM of DTT, 0.1 mg/mL of lysolecithin) per 10^8^ cells and collected (short spin mode, 10 s, 4 °C), and the supernatant was discarded. Then, the cells were resuspended in 333 μL of hypotonic buffer (20 mM of HEPES-KOH pH 7.5, 10 mM of KOAc, 1 mM of Mg(OAc)_2_, 4 mM of DTT) per 10^8^ cells. Then, the cell suspension was incubated on ice for 7 min and disrupted using a Dounce homogenizer (pestle B) for 20 strokes. The debris was pelleted (20,000× *g*, 10 min, 4 °C) and the supernatant was collected and stored in liquid nitrogen.

### 4.7. In Vitro Translation

Translation reactions were performed in a total volume of 10 μL, which contained 50% *v*/*v* S20 extract and translation buffer (10^x^ buffer consists of 10 mM of DTT, 5 mM of spermidine, 80 mM of creatine–phosphate, 10 mM of ATP, 2 mM of GTP, 100 mM of NaCl, 100 mM of potassium phosphate pH 7.5, 200 mM of HEPES-KOH pH 7.5, 150 complete amino acids mix (Promega, Madison, WI, USA), 100 mM of KOAc, and 50 ng of reporter mRNA). Reactions were conducted for 30 min at 30 °C, and luciferase expression was measured using a Luciferase Assay System kit (Promega, Madison, WI, USA).

### 4.8. Statistical Analysis

The data are plotted as boxes with Tukey-style whiskers for all mRNA transfection. All of the transfections have been replicated at least ten times. The statistical significance was determined using the two-tailed Mann–Whitney U test, as indicated in this article. All analyses were performed using GraphPad Prism 7. Outliers were excluded from the plots (but not from the analyses).

## Figures and Tables

**Figure 1 ijms-24-17149-f001:**
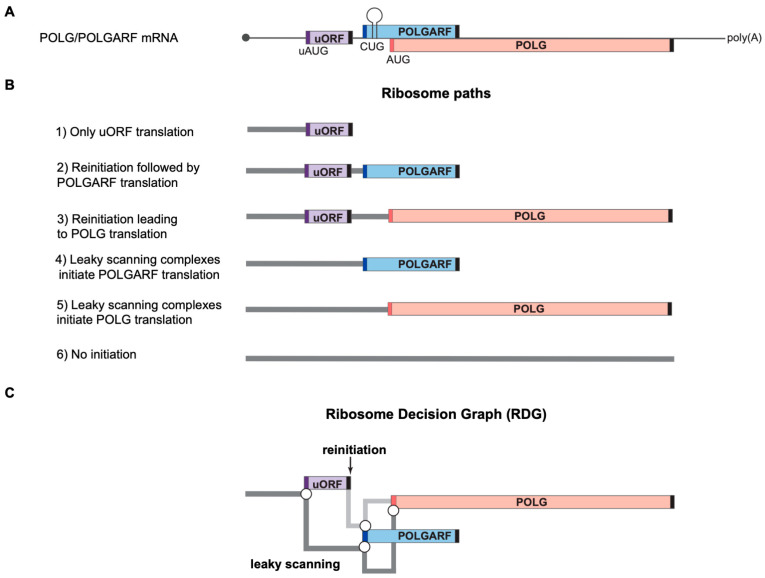
Ribosome paths to initiate POLG and POLGARF translation. Please note that the scheme has been drawn not to scale for readability. (**A**) A schematic representation of the natural dual-coding human POLG/POLGARF mRNA. The mRNA bears a ~300 nt long rather GC-rich (65% GC) 5′ UTR with a regulatory uORF located about 150 nt from the 5′ end and encoding 24 amino acids (the uORF is shown as a purple box, and dark purple bars depict the uAUG). There are only 11 nucleotides separating the uORF stop codon from the POLGARF start codon (the dark blue bar). The POLGARF ORF starts with the CUG codon in an unusually strong context, overlaps the main POLG ORF by more than 20%, and encodes a highly conserved ~25 kDa protein. The close-by stem-loop (referred to as SL in the other figures) ensures the efficient translation initiation on this CUG. The distance between the POLGARF start codon and the main POLG start codon (red bar) is only 53 nucleotides. Black bars depict stop codons. (**B**) Ribosome paths through the POLG/POLGARF mRNA. Reinitiation and leaky scanning are specified with respect to the uORF. All ribosomes that initiate translation on the main POLG start codon have leaked through the POLGARF start codon, regardless of their path through the uORF (leaky scanning or reinitiation). (**C**) Ribosome Decision Graph (RDG) representing translation as multiple ribosome paths through the POLG/POLGARF 5′ UTR. Boxes depict the ORFs and circles depict branching points where the ribosome makes a “decision” about whether it initiates or not. The path of leaky scanning complexes towards the downstream start codons is shown in dark grey. The light grey path represents the post-terminating small ribosome subunit that resumes scanning and can initiate on the downstream POLGARF or POLG start codons (the reinitiation path). The concept of visualization was adopted from [20].

**Figure 2 ijms-24-17149-f002:**
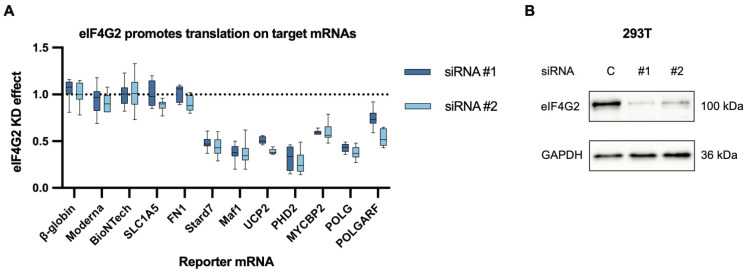
eIF4G2 participates in the translation of both POLGARF and POLG. (**A**) The contribution of eIF4G2 to translation was assessed using reporter mRNA transfection in 293T cells depleted of eIF4G2. Cells pretreated with either control or anti-eIF4G2 siRNAs for 72 h were transfected with in vitro transcribed m^7^G-capped and polyadenylated reporters. The Nluc-coding reference β-globin reporter mRNA was co-transfected with all of the reporters. The data are presented as ratios of normalized reporter expression in the eIF4G2-depleted to the control cells (the eIF4G2 KD effect). The knockdown effect < 1 corresponds to translation inhibition in the absence of eIF4G2. The translation of Stard7, Maf1, and UCP2 mRNAs has previously been shown to require eIF4G2. The reporter mRNAs with β-globin, SLC1A5, FN1, Moderna, and BioNTech 5′ UTR do not need eIF4G2 and serve as a negative control. For all eIF4G2 targets and β-globin reporters, the number of replicates exceeds 20. For the non-targets *n* ≥ 5. (**B**) Western blot analysis of the eIF4G2 knockdown in 293T cells, GAPDH as a loading control.

**Figure 3 ijms-24-17149-f003:**
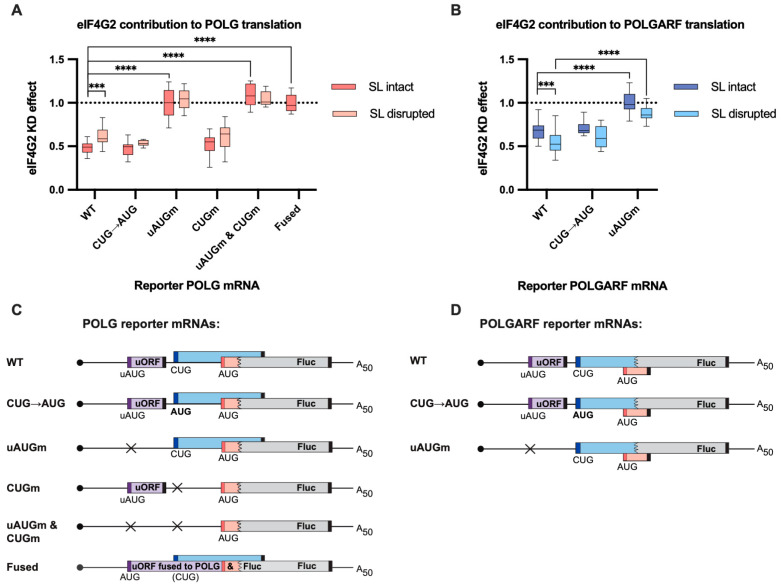
eIF4G2 promotes POLG and POLGARF translation. (**A**) In vitro transcribed m^7^G-capped and polyadenylated POLG reporters with the indicated wild-type (WT) or mutated 5′ UTRs were transfected into mock- and eIF4G2-depleted (siRNA#1) 293T cells along with the reference β-globin reporter mRNA (*n* ≥ 10). All assayed reporters were tested with either the wild-type or the disrupted stem-loop (referred to as SL intact or SL disrupted, respectively). The effect of the knockdown (eIF4G2 KD effect) is calculated by dividing the normalized reporter expression in eIF4G2-depleted cells by that in control cells. The knockdown effect < 1 reflects translation inhibition by the eIF4G2 depletion. The statistical significance is determined using the Mann–Whitney U test. Three and four asterisks stand for *p* < 0.001 and *p* < 0.0001, respectively. (**B**) Results of POLGARF reporter mRNA transfections (similar to panel A). (**C**) Schematic representation of the POLG reporters examined in panel A (not to scale). The POLG start codon (red bar) drives translation of the chimeric reporter protein, which consists of 15 N-terminal POLG amino acids (shown in pink) fused to the firefly luciferase (Fluc). The POLGARF CUG codon (the dark blue bar) drives the translation of an ORF that overlaps out of frame with the Fluc and encodes the 60-aa-long peptide with 33 N-terminal POLGARF amino acids (CUG-driven uORF is displayed in blue). The purple box depicts the regulatory uORF. The start codons are shown in corresponding color bars, and black bars depict the stop codons. The stem-loop is omitted for the sake of readability. Crosses display the positions of the upstream (with respect to the POLG ORF) start codons that were substituted for the stop codons in the corresponding reporters. In the reporter mRNA called “fused”, the uORF is fused in frame to the chimeric POLG/Fluc sequence. This resulting chimaera evaluates the role of eIF4G2 in the POLG/POLGARF mRNA translation initiation from the 5′-end to the uAUG. (**D**) Panel D is similar to panel C, with the exception that POLGARF reporter mRNAs are displayed. The POLGARF CUG start codon drives translation of a chimeric reporter protein, consisting of 33 N-terminal POLGARF amino acids (shown in blue) fused to the Fluc. The POLG start codon is out of frame with Fluc and provides translation of a 20-amino-acid-long stub (shown in pink).

**Figure 4 ijms-24-17149-f004:**
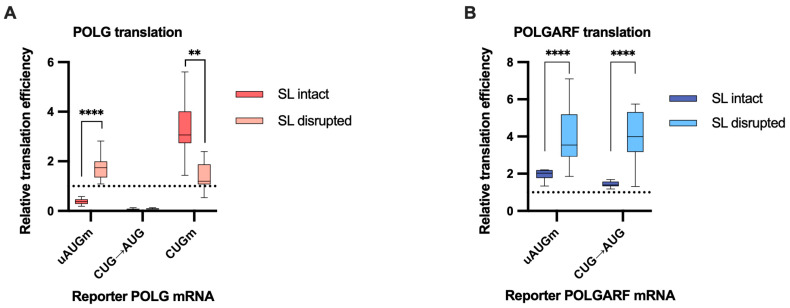
Mutating the start codons affects POLG and POLGARF translation efficiencies. m^7^G-capped and polyadenylated reporters with the indicated wild-type (WT) or mutated 5′ UTR were transfected into 293T cells along with the reference β-globin reporter mRNA (*n* ≥ 15). Nluc activity was used to normalize Fluc reporter expression. All assayed reporters were tested with either the wild-type or the disrupted stem-loop sequence (referred to as SL intact and SL disrupted, respectively). Relative translation efficiency is calculated by dividing the normalized expression of a mutant reporter (with the start codon altered) by the normalized expression of the wild-type reporter. When reporters with disrupted stem-loops are used, relative translation efficiency is shown with respect to the corresponding constructs with wild-type start codons and disrupted stem-loops. The dotted line at 1 corresponds to translation efficiency identical to that of the wild-type reporter mRNA. (**A**) Results of POLG reporter mRNA transfections. The statistical significance is determined using the Mann–Whitney U test. Four and two asterisks indicate *p* < 0.0001 and *p* < 0.01, respectively. (**B**) Results of POLGARF reporter mRNA transfections (similar to panel **A**). Four asterisks stand for *p* < 0.0001.

**Figure 5 ijms-24-17149-f005:**
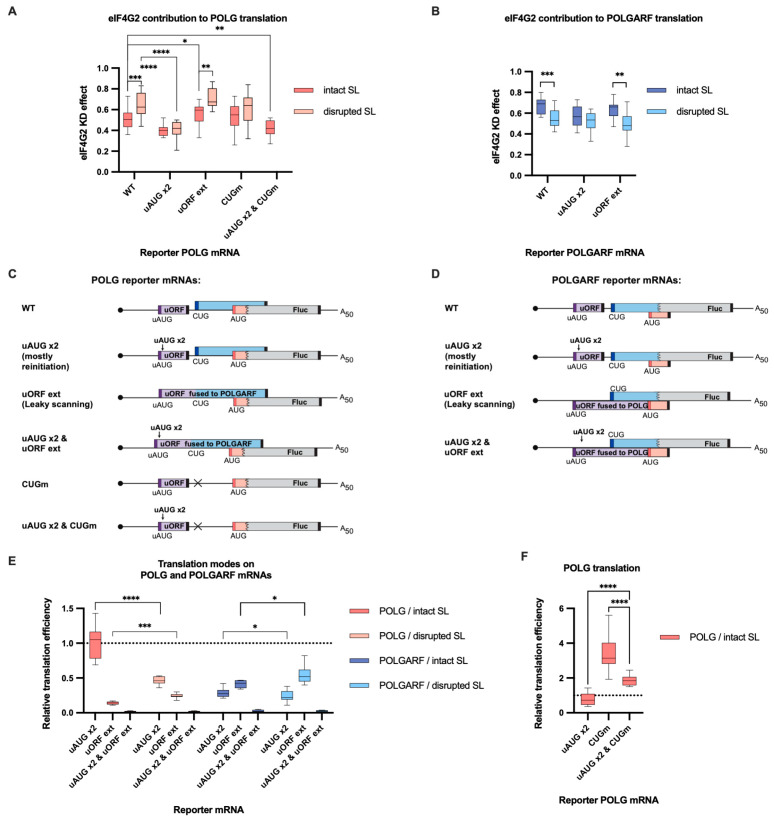
eIF4G2 promotes both leaky scanning and reinitiation on POLG and POLGARF mRNAs. (**A**) In vitro transcribed m^7^G-capped and polyadenylated POLG reporters with the indicated wild-type (WT) or mutated 5′ UTRs were transfected into mock- and eIF4G2-depleted 293T cells alongside reference β-globin reporter mRNA (*n* ≥ 10). The knockdown was accomplished using siRNA #1 against eIF4G2. Almost all assayed reporters were tested with either a wild-type or a disrupted stem-loop sequence (referred to as “SL intact” and “SL disrupted”, respectively). The knockdown effect (eIF4G2 KD effect) is calculated by dividing normalized reporter expression in eIF4G2-depleted cells by normalized expression in control cells. The knockdown effect < 1 signifies a translation inhibition in the absence of eIF4G2. The statistical significance is determined using the Mann–Whitney U test. The asterisks indicate *p*-value (one, two, three, and four asterisks stand for *p* < 0.05, *p* < 0.01, *p* < 0.001, and *p* < 0.001, respectively). (**B**) Results of POLGARF reporter mRNA transfections (similar to panel **A**). (**C**) Schematic representation of the POLG reporters examined in panel A (not to scale). Ribosomes can reach the main POLG start codon via leaky scanning through the uAUG or by reinitiation after completing the translation of the uORF. To estimate the eIF4G2 contribution to the leaky scanning, the uORF stop codon was mutated so that the extended uORF became fused to the POLGARF ORF and overlapped out of frame with the firefly luciferase ORF (referred to as “uORF ext”). To significantly reduce the leaky scanning through the uORF, two extra start AUG codons (referred to as “uAUG x2”) were inserted in frame and near the uAUG codon. On such a 5′ UTR, ribosomes mostly reach the main POLG start codon via reinitiation. Both mutations were introduced into the POLG 5′ UTR (referred to as “uAUG x2 & uORF ext”) to estimate the scanning complexes’ retention at the uORF in-frame start codons. The reinitiated scanning complexes on their way to the POLG start codon might interfere with the 80S ribosomes on the POLGARF ORF, thus increasing the need for eIF4G2. To estimate if this is the case, we compare the eIF4G2 contribution to reinitiation on the “POLG uAUG x2” reporter with that on the corresponding reporter mRNA with a mutated POLGARF start codon (referred to as “uAUG x2 & CUGm”). The start codons are shown in corresponding color bars, and black bars depict stop codons. (**D**) Panel D is similar to panel C, with the exception that POLGARF reporter mRNAs are displayed. The uORF stop codon was mutated so that the extended uORF became fused to the POLG ORF stub and overlapped out of frame with the firefly luciferase ORF (referred to as “uORF ext”). The POLGARF “uAUG x2” and “uAUG x2 & uORF ext” reporters are constructed similarly to corresponding POLG reporters. (**E**) Contribution of leaky scanning and reinitiation to translation of POLG and POLGARF mRNAs. Relative translation efficiency is calculated by dividing the normalized expression of a mutant reporter (designated as “uAUG x2” or “uORF ext” to address leaky scanning and reinitiation, respectively) by the normalized expression of a wild-type reporter. When reporters with disrupted stem-loops were used, the relative translation efficiency is shown with respect to corresponding reporters having wild-type start codons and a disrupted stem-loop. The dotted line at 1 corresponds to translation efficiency identical to that of the wild-type reporter mRNA. Note that the contribution of the reinitiation mode can be estimated through the subtraction of leaky scanning mode from 1. The “uAUG x2” reporter constructs arguably reflect the upper estimate for the contribution of reinitiation mode on the wild-type mRNA. The statistical significance was determined using the Mann–Whitney U test. The asterisks indicate *p*-value (four, three, and one asterisks stand for *p* < 0.0001, *p* < 0.001, and *p* < 0.01, respectively). (**F**) Reinitiating ribosomes mainly bypass the POLGARF CUG start codon. Relative translation efficiency is calculated in the same way as in Panel E, and statistical significance is determined similarly. Asterisks denote *p* < 0.0001.

**Figure 6 ijms-24-17149-f006:**
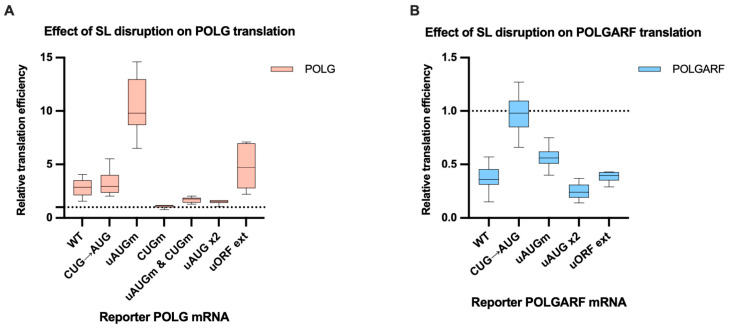
The disruption of the stem-loop affects the translation efficiency of POLG and POLGARF mRNAs. m^7^G-capped and polyadenylated reporters with the indicated wild-type (WT) or mutated 5′ UTR were transfected into 293T cells along with the reference β-globin reporter mRNA (*n* ≥ 15). Nluc activity was used to normalize Fluc reporter expression. All assayed reporters were tested with either a wild-type or a disrupted stem-loop sequence (SL disruption). The effect of the stem-loop disruption is calculated by dividing the normalized expression of a mutant reporter with the disrupted stem-loop by the normalized expression of a corresponding reporter with the intact stem-loop. The dotted line at 1 corresponds to translation efficiency identical to that of the wild-type reporter mRNA. (**A**) Results of reporter POLG mRNA transfections. (**B**) Results of reporter POLGARF mRNA transfections (similar to panel (**A**)).

**Figure 7 ijms-24-17149-f007:**
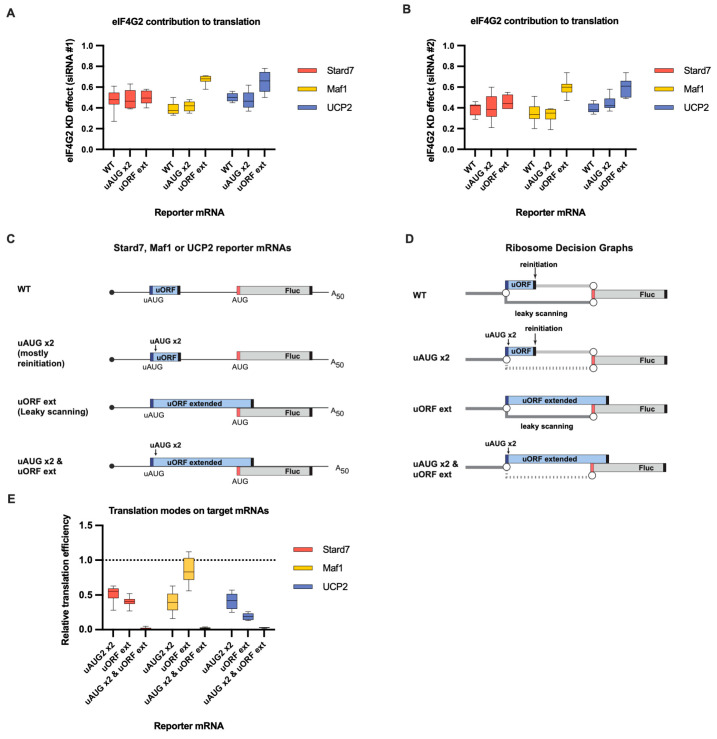
eIF4G2 promotes both leaky scanning and reinitiation on Stard7, Maf1, and UCP2 mRNAs. (**A**) In vitro transcribed m^7^G-capped and polyadenylated Maf1, Stard7, and UCP2 reporters with the indicated wild-type (WT) or mutated 5′ UTRs were transfected into mock- and eIF4G2-depleted (siRNA#1) 293T cells alongside reference β-globin reporter mRNA (*n* ≥ 10). The effect of the knockdown (eIF4G2 KD effect) is calculated by dividing normalized reporter expression in eIF4G2-depleted cells by normalized expression in control cells. The knockdown effect <1 shows translation inhibition in the absence of eIF4G2. (**B**) Panel B is similar to panel A, except siRNA #2 was used for the eIF4G2 depletion in 293T cells *(n* ≥ 10). (**C**) Schematic representation of the reporters examined in panels A, B, and E (not in scale). Stard7, Maf1, and UCP2 5′UTRs contain uORFs (shown in blue box). Dark blue and red bars depict uAUG and main start codons, respectively, and black bars depict stop codons. All analyzed 5′ UTRs were mutated to address leaky scanning and reinitiation separately. Leaky scanning is addressed using an mRNA reporter with an extended uORF that overlaps significantly with the Fluc coding sequence (referred to as “uORF ext”). The insertion of two additional uAUGs into uORF (designated as “uAUG x2”) makes the reinitiation nearly the only way for ribosomes to reach the main start codon. To estimate the scanning complexes retention at the uORF in-frame start codons, both mutations were introduced into these 5′ UTRs (referred to as “uAUG x2 & uORF ext”). (**D**) Ribosome Decision Graphs representing translation as multiple ribosome paths through the wild-type or mutated 5′ UTRs. Boxes demonstrate ORFs. Circles depict branching points where the ribosome makes a “decision” of whether to initiate or not. The path of leaky scanning complexes towards the downstream start codon is shown in dark grey. Light grey paths represent the post-terminating small ribosome subunit that resumes scanning and can initiate on the main start codon (reinitiation path). The dotted paths represent the significant decrease in the corresponding mode of initiation. (**E**) The contributions of leaky scanning and reinitiation to the translation of Stard7, Maf1, and UCP2 mRNAs. Relative translation efficiency is calculated by dividing the normalized expression of a mutant reporter (designated as “uAUG x2” or “uORF ext” to address leaky scanning and reinitiation, respectively) by the normalized expression of a wild-type reporter (*n* ≥ 10). The dotted line at 1 corresponds to translation efficiency identical to that of the wild-type reporter mRNA. Note that the contribution of the reinitiation mode can be estimated through subtraction of the leaky scanning mode from 1. The “uAUG x2” reporter constructs reflect the upper estimate for the contribution of the reinitiation mode on the wild-type mRNA.

**Figure 8 ijms-24-17149-f008:**
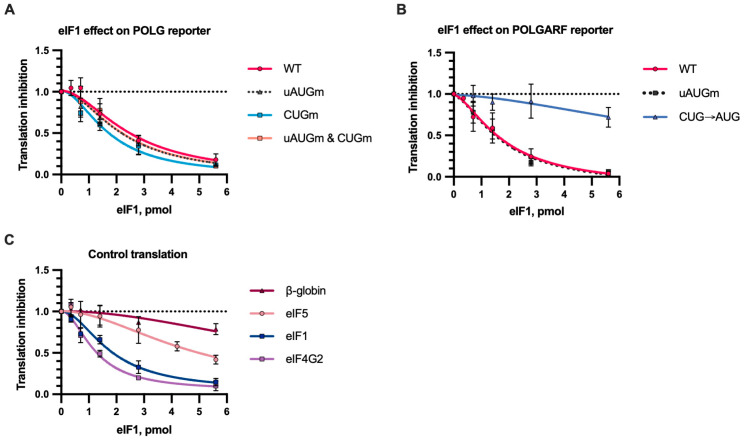
In vitro translation with eIF1. Reporter mRNAs were translated in S20 cell extract from Expi293F cells in the presence of eIF1 or a storage buffer (*n* ≥ 5). The Fluc activity (or Nluc in the case of the eIF1 reporter) was measured. Translation inhibition by eIF1 is demonstrated relative to reporter activity in the presence of a storage buffer. The dotted line at 1 represents translation that is completely unresponsive to eIF1. (**A**) eIF1 inhibits translation of reporter POLG mRNAs. (**B**) The effect of eIF1 on the translation of POLGARF reporter mRNAs. (**C**) The effect of eIF1 on the translation of β-globin, eIF5, eIF1, and eIF4G2 reporter mRNAs. The reporter mRNA for eIF1 and eIF4G2 has wild-type GUG start codons.

**Figure 9 ijms-24-17149-f009:**
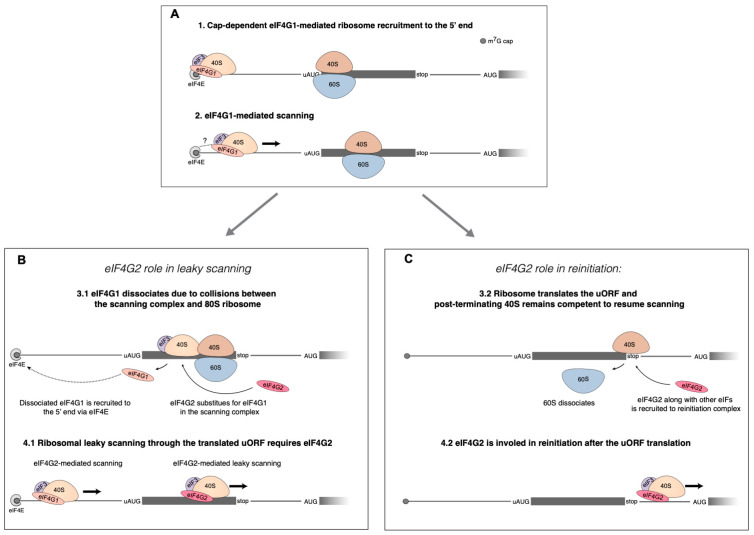
The proposed eIF4G2′s roles in cap-dependent translation of mRNA with uORFs. (**A**) The initial steps of translation initiation occur in accordance with the conventional Kozak’s mechanism. The small ribosome subunit is recruited to the 5′ end of mRNA via interaction between m^7^G-cap and eIF4F. Then eIF4G1 facilitates scanning from the very 5′ end. The 80S ribosome is depicted translating the uORF. uORF translation leads to the need for eIF4G2 that can mediate both leaky scanning and reinitiation on a single mRNA at the same time. (**B**) The role of eIF4G2 in leaky scanning has been linked to a partial loss of eIF4G1 caused by collisions of the scanning complex and the 80S ribosome within the uORF. Through interaction with eIF4E, the eIF4G1 gets back to the m^7^G-cap, and the role of helicase is then assumed by the eIF4G2–eIF4A complex, thereby mediating leaky scanning. (**C**) eIF4G2 can also facilitate reinitiation on the same mRNA. eIF4G2 assists the post-terminating 80S ribosome to resume scanning and thus contributes to reinitiation. The scheme from [5] was elaborated.

## Data Availability

All of the data presented in this study are available upon request from the last author.

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
