# Peer review of "The Roles of eIF4G2 in Leaky Scanning and Reinitiation on the Human Dual-Coding POLG mRNA"

_ijms, 2023, doi:10.3390/ijms242417149_

Round 1

Reviewer 1 Report

Comments and Suggestions for Authors

Shestakova and others present a manuscript dissecting possible function of eIF4G2 in translation on meseenger(m)RNAs that possess upstream open reading frames (uORFs). They use POLG mRNA as it offers a convenient interrogatory system, allowing to discriminate between its different main ORFs, the shorter and more 5' proximal POLGARF and the longer and starting downstream POLG. These ORFs have a partial overlap.

Overall the results are presented with detail and manuscript well-written, with some minor typos/inconsistencies (some included in the minor points below), the logic of the investigation is presented well on the most part, and the experiments with inhibitors, knock-down and variants of the constructs look all convincing. Some text portions seem misplaced though, see in major and minor points.

Major points

  1. The authors juxtapose leaky scanning and re-initiation (and then combine these back), but can these really be juxtaposed? These mechanisms, generally, can be independent one on another. Thus, they can co-occur probabilistically, with some initiation events following one and then some the other path. I think this slightly artificial juxtaposition can be removed.

  2. The authors rely, to a large extent, on in vitro systems. This gives some freedom in experimental design, but also imposes responsibility on accuracy of the techniques and conclusions. Of a particular concern for me was a relatively small volume of the in vitro translation reaction (10 microliters) and a fixed incubation time of 30 minutes. Could some additional experiments be provided for the critically-observed differences in translation that would use larger volume (20 microliters, for example) and take the dynamics of signal accumulation. A concern here is that at 30 minutes translation reactions may have reached their saturation and thus the authors are unable to isolate true differences in translational outputs. Ideally, some information on the intactness of the mRNA at the corresponding points could be useful (I understand the latter is an additional experiment and thus do not insist it should be done, but it could elevate pitch of the work).

  3. Another concern is that in all of the experiments, mRNA is being presented to the system in a non-natural form, and effectively what is observed are some initial rounds of translation, not necessary steady-state translation as it occurs in the cells. This is to do with the mRNP packaging and behaviour of in vitro translated or transfected mRNA. Could the authors add an experiment that detects translation of endogenous mRNA? I think this can be possible relatively straightforwardly, for example by ribo-seq on knock-down vs. naive cells. This would improve the work greatly, as it can be argued that the effects currently observed are somewhat artificial. It also would provide an orthogonal validation of the results.

  4. Another concern is the use of highly artificial reporter constructs, which have a long luciferase ORF fused with the ORFs of interest (of what length?). There is little guarantee these constructs would perform alike to the “original wild type” arrangement. This is another reason to consider an orthogonal validation.

  5. The manuscript must be supplemented with schematics depicting the proposed models in consideration. Ideally, these models need to be also discussed graphically in the context of all or some of the tested constructs.

  6. Figure 3A,B, Figure 4A,B, Figure 5A,B,E,F and alike, please indicate exactly what measurement Y-axis depicts? This also must be explained very clearly in the figure legend.

  7. The logic of interpreting various construct alterations, such as stem loop abrogation, start codon editing etc. must be presented better. It currently is convoluted, and together with the absence of any graphical interpretation, is difficult to follow.

  8. What is the proof that “POLGARF translation became completely dependent on the ribosomes that leak through the uAUG” when extending uORF (lines 187-191)? This could still be initiated internally?

  9. Lines 316-342, all of the eIF1 part looks a bit out of context. Is it necessary? If the authors wish to retain it, I think it needs to be presented and explained better. Perhaps, something in the introduction could be also said about start codon selection stringency and uORF leak/scan through effects in conjunction with the eIF1 function. Overall conclusion that eIF1 will have different effect depending on the nucleotide and mechanistic context is nice, but a bit obvious.

  10. Line 346, please either remove “historic” reference, or appropriately explain it. The manuscript is about certain mechanisms and not history of the study. Overall I think Discussion is difficult to follow. Can it be better structured: by mechanisms, by mRNA type, or by experimental approach? It definitely misses illustrations, too.

Minor points

  1. Line 17, it can be argued eIF4G2 is not poorly studied. Perhaps, insufficiency.

  2. Line 19, I would remove “unique”. In a sense, every mRNA is unique.

  3. Line 34, “the initiation cycle, it engages” Unclear what “it” means.

  4. Lines 56-62, perhaps either mention much earlier (but the current text flow does not allow it) or in the discussions/outlook. Current location breaks the logic flow in the introduction. Instead of this text, it is better to write a transition to the POLG mRNA – currently it is a bit abrupt. Maybe to put forward a justification of the choice?

  5. Line 77, Figure 1 schematic – it would be useful to indicate lengths of the regions of the mRNA, and maybe some other basic information such as GC content.

  6. Lines 116-132, most of this info should be in the introduction!

  7. Line 153, Figure 3 – please add lengths of the mRNA regions.

  8. Line 428, “3 mM (Biosan, Novosibirsk)”. 3 mM of what? NTPs each?

  9. Line 433, “RNA was precipitated by LiCl on ice for one hour.”. How LiCl was added?

  10. Line 497, “The APC was funded by XXX.” Needs correction?

  11. Western blots look convincing.

Comments on the Quality of English Language

Minor editing is required. Some non-exhaustive examples are mentioned in the minor points.

Author Response

Thank you for the positive evaluation of our work.

What we've done is address the reviewers' points and, although we weren't asked, add data from another cell line, Huh7. They perfectly reproduce what was previously shown in 293T cells and all of this is now in supplementary figures.

Major points

  1. You are right, but that was an integral part of the plot. We feel that many people see reinitiation and leaky scanning as almost mutually exclusive processes. If the context is good, then you have reinitiation, if the context is bad, then you have leaky scanning. In part, this reflects the evolution of our own understanding of uORFs. 

  1. We do not fully understand the expressed concern. Only the in vitro translation reactions were incubated for 30 minutes, while the majority of presented results were obtained via mRNA transfection into cultured cells. Addressing translation inhibition by eIF1, we did assay the kinetics of translation and we can assure you that the reactions do not reach the saturation and there is still a linear accumulation of the product at 30 minutes. On the other hand, 95% of the presented data have been obtained from transfected cells. We acknowledge the fact that cultured cells are often referred to as in vitro, but we do not feel this is what you mean.

Since northern blots or RT-qPCR are not applicable to evaluate stability of the lipofected RNA, we routinely perform kinetics analysis of transfected mRNAs expression. The idea behind this is that accumulation of Fluc protein slows down for a less stable mRNA compared to that for a more stable mRNA. All the investigated mRNAs are translated with similar kinetics, thus there is little difference between their stabilities.

  1. While this notion may be perfectly relevant to in vitro translation, why should this be true for translation in cells is less clear. Again, we routinely perform kinetic assays, and 3-4 hours post transfection is a timepoint when the luciferase is accumulating with no sign of a decline. Examples can be found in our previous articles. Using ribosomal profiling is a good idea, but, both in our data and in the published eIF4G2-depletion ribosome footprint profilings, the coverage is not good enough to make a statistically significant statement.

  1. We agree that using orthogonal approaches strengthens any investigation, but we cannot imagine another assay that could let us unambiguously distinguish between the reinitiation and the leaky scanning. 33 and 15 N-terminal amino acids were fused to Fluc (550 aa long) in the case of POLGARF and POLG reporters, respectively, in order to address the POLG and POLGARF translation separately. 

  1. We agree that mechanistic schemes were missing, and we have added them in the form of brand new ribosome decision graphs (Tierney et al., 2023).

  1. What measurement the Y-axis shows is what effect on the reporters' translation has an eIF4G2 knockdown. We have slightly expanded the figures' legends. 

  1. We have added certain linking sentences and graphical explanations.

  1. We tend to think that no noticeable internal initiation occurs here. When we test the reporters that can only be translated via leaky scanning (the uORF overlaps with the FLuc sequence), but no leaky scanning is technically possible due to the introduced AUG codons that "catch" the leaky ribosomes, we see that the efficiency of translation of such mRNAs is two orders of magnitude lower than that of the wild-type reporters. Even if this reflects the internal initiation, the level is too low to take it into account.

  1. Perhaps not exactly exciting new knowledge, but we have this piece of data and we think there is no reason not to publish it, not least because we do not plan to continue this study.

  1. We have slightly rewritten the Discussion and have added subsections. The diagram is now there, just as suggested.

Minor points

  1. Yes, eIF4G2 has been studied quite a lot, but the outcome is we poorly know how it functions. But we agree that the initial wording is an exaggeration, so we've changed it as suggested.
  2. We agree all mRNAs are unique, but some mRNAs are more unique than others. Cases like the POLG are extremely rare, so we believe saying the POLG mRNA is unique is fully justified.
  3. Explained.
  4. We removed the last sentence of the preceding paragraph to make the flow more fluent. However, we feel that otherwise the logic is fine. We discuss what eIF4G2 depletion messes up in the cell and come to mitochondrial disfunction. At this point we introduce the POLG, mitochondrial DNA polymerase. 
  5. We have greatly expanded the figure. First, we included two panels that explain the ways of the translation of POLG and POLGARF cistrons to make understanding of the following text easier. Second, we also expanded the figure legend to address the reviewer's question.
  6. A part of the paragraph has been relocated to the Introduction as suggested.
  7. This information is now included in the legend for Figure 1.
  8. Yes, 3 mM each. Now explained.
  9. The concentrations are now included, thank you for the note.
  10. Yes, we missed this, now fixed, thank you.
  11. Uhm, the blots are genuine, I dare say.

Reviewer 2 Report

Comments and Suggestions for Authors

Overall this is an expertly present study of quite some interest to those in the translation field. The figures are generally well-prepared and clear, although I think that some of the legends and/or the accompanying results text are not sufficiently detailed to appreciate all that is shown. Application of statistical testing (asterisks) is inconsistent across the figure panels. Can this be done for all data? Why are 'error bars' shown according to Tukey and not SD or SEM? Some figures contain data with many more mRNAs (positive and negative controls!?) than is explained. Figure 2A is one such case, as is Figure A1. It is not clear at all what the 'PPP242 effect' parameter really is in Figure A1, and related to that I don't understand what the dashed line represents. Figure 8 legend does not state the number of replicates.

The choice of cell line for transfection and for preparation of the in vitro translation extracts could be better explained, both in the methods and in the results sections.

Finally, can the authors comment on the possible effect of mRNA stability on their results? Can this be excluded as a contributing factor?

Comments on the Quality of English Language

The manuscript is written clearly enough to be understandable. There are some minor grammatical errors that could be fixed in revision.

Author Response

Thank you for the sympathetic review.

What we've done is address the reviewers' points and, although we weren't asked, add data from another cell line, Huh7. They perfectly reproduce what was previously shown in 293T cells and all of this is now in supplementary figures.

Specifically:

  1. We have expanded the legends a bit to make them more detailed, as suggested.
  2. There are two reasons for our choice of data presentation. First, the number of experimental points does not allow us to assume a normal distribution, whereas the SD/SM strictly require one. The interquartile range (IQR) is calculated directly from the data regardless of the distribution, so we use Tukey style with boxes representing the IQR and whiskers corresponding to 1.5xIQR. Second, the boxes provide the reader with somewhat more information (see e.g. Krzywinski, M., Altman, N. Visualizing samples with box plots. Nat Methods 11, 119-120 (2014). https://doi.org/10.1038/nmeth.2813). We also use non-parametric tests, such as the Mann-Whitney U test, which do not require the data to follow a normal distribution. 

  1. The missing statistical tests (the asterisks) have been added.

  1. The dotted line always shows a lack of effect, which is why it always corresponds to "one" on the Y-axis. For the PP242 or thapsigargin treatments, it's a bit more complicated. Firstly, the translation of the reference (β-globin-Nluc) also drops and the "one" shows the effect identical to that of the reference and you can see that the β-globin-Fluc reporter always hovers around the dotted line. Secondly, there is another line, the dashed line, which shows the average effect of a treatment on the reference β-globin-Nluc reporter. We routinely treat cells with a drug or vehicle and then apply the same transfection mixture to the treated and untreated cells, which ensures that the same amount of mRNA is applied to both. Thus, with dozens and dozens of experimental points, we obtain a statistic of how the particular treatment inhibits the reference mRNA. These data pass the normality test, by the way.. The dashed line therefore shows the inverse of this averaged inhibition, so that the position of a point below it reflects an inhibition and that above it a stimulation in terms of absolute values. It can be seen that the PTV IRES not only withstands the mTOR treatment, but actually starts to function better. You can also see that the ATF4 reporter ignores eIF2 phosphorylation after thapsigargin treatment, but only ignores because it isn't translated better. This is explained clearly in the figure legend now.

  1. The number of replicates for Figure 8 has been added to the legend.

  1. mRNAs used as negative and positive controls are now indicated in the legends.

  1. The majority of presented results were obtained by reporter mRNA transfection into adherent 293T and Huh7 cells. The preparation of cell extract for in vitro translation requires 3-6 billion cells, thus we used easily grown Expi293F suspension cells rather than 293T adherent cells to obtain S20 translation extract. This concern is now added in the methods section.

  1. Since northern blots or RT-qPCR are not applicable to evaluate stability of the lipofected RNA, we routinely perform kinetics analysis of transfected mRNAs expression. The idea behind this is that accumulation of Fluc protein slows down for a less stable mRNA compared to that for a more stable mRNA. All the investigated mRNAs are translated with similar kinetics, so there is little difference between their stabilities.